# CEP55 as a Promising Immune Intervention Marker to Regulate Tumor Progression: A Pan-Cancer Analysis with Experimental Verification

**DOI:** 10.3390/cells12202457

**Published:** 2023-10-15

**Authors:** Gang Wang, Bo Chen, Yue Su, Na Qu, Duanfang Zhou, Weiying Zhou

**Affiliations:** 1Department of Pharmacology, College of Pharmacy, Chongqing Medical University, Chongqing 400016, China; 2Chongqing Key Laboratory of Drug Metabolism, Chongqing Medical University, Chongqing 400016, China; 3Key Laboratory for Biochemistry and Molecular Pharmacology of Chongqing, Chongqing Medical University, Chongqing 400016, China

**Keywords:** pan-cancer, centrosomal proteins 55, tumor progression, immune infiltration, CD-437

## Abstract

CEP55, a member of the centrosomal protein family, affects cell mitosis and promotes the progression of several malignancies. However, the relationship between CEP55 expression levels and prognosis, as well as their role in cancer progression and immune infiltration in different cancer types, remains unclear. We used a combined form of several databases to validate the expression of CEP55 in pan-cancer and its association with immune infiltration, and we further screened its targeted inhibitors with CEP55. Our results showed the expression of CEP55 was significantly higher in most tumors than in the corresponding normal tissues, and it correlated with the pathological grade and age of the patients and affected the prognosis. In breast cancer cells, CEP55 knockdown significantly decreased cell survival, proliferation, and migration, while overexpression of CEP55 significantly promoted breast cancer cell proliferation and migration. Moreover, CEP55 expression was positively correlated with immune cell infiltration, immune checkpoints, and immune-related genes in the tumor microenvironment. CD-437 was screened as a potential CEP55-targeted small-molecule compound inhibitor. In conclusion, our study highlights the prognostic value of CEP55 in cancer and further provides a potential target selection for CEP55 as a potential target for intervention in tumor immune infiltration and related immune genes.

## 1. Introduction

Cancer threatens human health and is the second-leading cause of death worldwide [1]. Moreover, the number of cancer cases worldwide may increase by 60% in the next two decades, further contributing to one in six cancer deaths per year [2]. Current widely used cancer treatments include surgery, radiation therapy, chemotherapy, biologic therapy, and targeted therapy, all of which have shown some efficacy [3,4]. Unfortunately, the prognosis and survival of patients still cannot be effectively improved due to drug resistance, toxic side effects, low immunity, and compliance [5]. In recent years, immunotherapy has received attention from researchers, and the exploration of cancer immunotherapy has become a prominent avenue for cancer treatment, especially the development of immune biomarkers [6]. Therefore, it is necessary to search for and validate promising tumor immune biomarkers.

Tumors are usually caused by aberrant differentiation due to chromosomal genomic instability during cell division [7,8]. CEP55, a member of the centrosomal protein family [9], has been shown to be highly expressed in many human tumor tissues and correlates with tumor malignancy, invasiveness, and poor prognosis [10]. High levels of CEP55 form a complex with PI3K to activate PI3K/AKT activity, thereby enhancing the non-anchored growth of hepatocellular carcinoma (HCC) [11]. Knocking down CEP55 levels delayed epithelial-mesenchymal transition (EMT) and increased cisplatin sensitivity in breast cancer (BC) by inhibiting the P38MAPK and ERK1/2 pathways [12]. Moreover, abnormal expression of CEP55 significantly promoted endometrial cancer (EC) progression, whereas down-regulation of CEP55 expression could inhibit proliferation, invasion, migration, delay the cell cycle, and accelerate apoptosis [13]. These results suggest that CEP55 is a promising clinical target for cancer therapy. However, current studies are limited to a few types of cancer, and the therapeutic effects and mechanisms of CEP55 in various types of tumors are still not fully understood.

With the development of tumor molecular biology and genomics, more tumor molecular phenotypes have been revealed. This requires further development of targeted therapies for driver mutations to improve the cancer treatment system. In this study, through multiple databases, we found that the expression levels of CEP55 were significantly higher in most of the tumors than in the corresponding normal tissues, and they correlated with the pathological grade and age of the patients and affected the prognosis. Moreover, CEP55 expression was positively correlated with immune cell infiltration, immune checkpoints, and immune-related genes in the tumor microenvironment. Our results suggest a broad role for CEP55 in the diagnosis, prognosis, and immunotherapy of cancer, providing ideas for a comprehensive understanding of CEP55 in immunotherapy and the development of novel targeted therapies.

## 2. Materials and Methods

### 2.1. Data Collection

Samples of 33 cancer types containing clinical follow-up survival and staging information were obtained from The Cancer Genome Atlas (TCGA, https://www.cancer.gov/aboutnci/organization/ccg/research/structural-genomics/tcga accessed on 2 December 2022) database and converted to TPM format. Gene expression data for different tissues were obtained from the Genotype-Tissue Expression database (GTEx, https://gtexportal.org/ accessed on 2 December 2022) and the Cancer Cell Line Encyclopedia database (CCLE, https://portals.broadinstitute.org/ccle/about accessed on 2 December 2022). In all downloaded data, differentially expressed genes (DEGs) between tumor tissue and adjacent tissue were determined using log2 transformations and t-tests with *p*-values < 0.05. Data analysis was conducted using R software (Version 4.0.2; https://www.Rproject.org accessed on 28 December 2022). The cancerous tissues and corresponding normal tissues are shown in Appendix A.

### 2.2. Data Processing and Analysis

We downloaded the distribution of CEP55 in human tissues and some immunohistochemical images of CEP55 in tumor tissues and corresponding normal tissues through the Human Protein Atlas (HPA, https://www.proteinatlas.org/ accessed on 3 December 2022) website. The full extent of tumor immune cell infiltration, immune cell abundance, and CEP55 correlation with related genes were obtained from the Tumor Immune Estimation Resource2 (TIMER2, https://cistrome.shinyapps.io/timer/ accessed on 5 January 2023) website and TISIDB (http://cis.hku.hk/TISIDB/index.php accessed on 8 January 2023). The protein interaction network of CEP55 was predicted using the STRING database (https://string-db.org/cgi/input.pl accessed on 16 January 2023). Gene Oncology (GO) and the Kyoto Encyclopedia of Genes and Genomes (KEGG) analyzed molecules with potential roles in CEP55 using the clusterProfiler package. The Kaplan–Meier Plotter was used to analyze the correlation between CEP55 expression and patient survival in different cancers, while the results of univariate Cox regression were presented in a forest-like plot. For CEP55 protein expression, methylation, and phosphorylation status, we analyzed different tumor proteome datasets in the Clinical Proteomic Tumor Analysis Consortium (CPTAC, https://gdc.cancer.gov/about-gdc/contributed-genomic-data-cancer-research/clinical-proteomic-tumor-analysis-consortium-cptac accessed on 3 December 2022) using the University of Alabama at Birmingham Cancer data analysis Portal (UALCAN, https://UALCAN.path.uab.edu/index.html accessed on 3 December 2022) online. Copy number alterations (CNAs) and mutation types of CEP55 in pan-cancer patients were determined by the cBioPortal for Cancer Genomics (http://www.cbioportal.org/ accessed on 26 December 2022). We searched and obtained data related to CEP55 in clinically relevant alternative splicing (AS) events from the OncoSplicing database (http://www.oncosplicing.com/ accessed on 9 December 2022). Data to validate CEP55 expression and function at the single cell level were obtained from the free Tumor Immune Single-cell Hub (TISCH, http://tisch.compgenomics.org/home/ accessed on 1 January 2023) data platform. Differences in CEP55 expression between responders and non-responders and the receiver operating characteristic curves (ROC) for treatment-related survivors were plotted by searching on ROC plots [14] (www.rocplot.org accessed on 5 January 2023). Comprehensive resources for compounds associated with CEP55 expression levels and drug sensitivity were downloaded from cMap (https://clue.io/ accessed on 5 January 2023) and RNAactDrug (http://bio-bigdata.hrbmu.edu.cn/RNAactDrug/ accessed on 5 January 2023). For protein-compound interactions, we followed the previous approach [15,16].

### 2.3. Cell Culture and Transfection

Breast cancer cell lines MDA-MB-231, CAL-148, and SK-BR-3 were obtained from the American Type Culture Collection (ATCC: Manassas, VA, USA). MDA-MB-231 and CAL-148 were cultured in Dulbecco’s modified eagle medium (DMEM), and SK-BR-3 was cultured in 1640 containing 10% fetal bovine serum (FBS) at 37 °C in 5% CO_2_.

The human target gene CEP55 small interfering RNA (siRNA) was purchased from Tsingke Co., Ltd. (Beijing, China). The sequence of siRNA CEP55-1 was 5′-GCCUGAAUCAGAAGGUUAU-3′ and siRNA CEP55-2 was 5′-GCAGCAUCAAUUGCAUGUA-3′. Expression of CEPP5 was silenced by CEP55 siRNA with Lipofectamine 8000 (Beyotime, Shanghai, China) under the guidance of the manufacturer’s instructions. To overexpress CEP55, we transiently transfected the CEP55 plasmid (Sino Biological, Beijing, China) into tumor cells with an empty vector as a negative control, following the method mentioned above.

### 2.4. RNA Extraction and qRT-PCR

Total RNA was extracted from knockdown siRNA CEP55-treated MDA-MB-231 and CAL-148 cells using TRIzol reagent (Invitrogen, Carlsbad, CA, USA) according to the manufacturer’s protocol. Subsequently, cDNA synthesis and real-time polymerase chain reaction (qPCR) were performed with the PrimeScript RT kit and SYBR Green Premix Ex Taq (TaKaRa, Dalian, China), respectively. GAPDH was used for normalization, and mRNA expression was evaluated by the comparative CT (2^−ΔΔCT^) method. All primers used for RT-qPCR are as follows: CEP55 forward primer: 5′-TGAAGAGAAAGACGTATTGAAACAA-3′; CEP55 reverse primer: 5′-ACTGTGGCTCCAAACTGCTT-3′; GAPDH forward primer: 5′-GAATGGGCAGCCGTTAGGAA-3′; GAPDH reverse primer: 5′-GAGGGATCTCGCTCCTGGAA-3′.

### 2.5. Western Blot Analysis

The breast cancer cell lines MDA-MB-231 and CAL-148 were transfected with the corresponding siRNA, CEP55, for 48 h. Cells were lysed with RIPA buffer containing a mixture of protease inhibitors and phosphatase inhibitors (Beyotime, Shanghai, China). After quantification and denaturation for each protein sample, 30 μg of protein was separated and transferred onto polyvinylidene difluoride (PVDF) membranes (Millipore, Burlington, MA, USA) by sodium dodecyl sulfate polyacrylamide gel electrophoresis (SDS-PAGE). Membranes were exposed to a 5% skim milk solution for 2 h at room temperature, incubated with primary antibody (Cell Signaling Technology, Beverly, MA, USA) overnight at 4 °C, then reacted with HRP-coupled secondary antibody for 2 h and developed using ECL substrate. Mouse monoclonal anti-GAPDH (Cell Signaling Technology, Beverly, MA, USA) was used as an internal standard.

### 2.6. Cell Viability

To determine the effect of knockdown of CEP55 on cells, cell viability was measured using a 3-(4,5)-dimethylthiahiazo (-z-y1)-3,5-di-phenytetrazoliumromide (MTT) assay according to the manufacturer’s instructions (Beyotime, Shanghai, China). MDA-MB-231 and CAL-148 cells were inoculated in 96-well culture plates (3 × 103 cells/well) and intervened with siRNA CEP55, then placed in an incubator at 37 °C with 5% CO_2_ for 24 h, 48 h, 72 h, and 96 h. After that, the absorbance was measured at 570 nm with a Spectra Max M5 microplate reader (Molecular Devices, Sunnyvale, CA, USA). The data represent the mean ± SD of three independent experiments with five replicates in each experiment.

### 2.7. Cell Proliferation and Migration Assays

Clone Formation Assay: MDA-MB-231 and CAL-148 cells from the si-NC and si-CEP55 groups were seeded on 12-well plates (500 cells/well) and cultured in a 5% CO_2_ incubator at 37 °C for at least 2 weeks. The colonies formed were fixed in 4% methanol and later stained with crystalline violet dye, photographed, and counted. 5-Ethynyl-2′-deoxyuridine (EdU) assay (Beyotime, Shanghai, China): Treated MDA-MB-231 and CAL-148 cells in 48-well plates were incubated with the recommended concentration of EdU for 2 h, and then the click reaction was performed according to the manufacturer’s instructions. Images were collected using a fluorescence microscope (Nikon, Ni-U, Tokyo, Japan). Transwell: MDA-MB-231 and CAL-148 cells treated with si-CEP55 for 48 h were suspended in the upper chamber of each well with DMEM (5 × 104 cells/well), and medium containing 10% FBS was added to the lower chamber. After 48 h of incubation, the non-migrating cells remaining on the top surface were gently removed with a cotton swab, and the migrating cells were fixed, stained with crystal violet, and counted under a light microscope. Wound healing: All cells were inoculated in 6-well plates, and a linear wound was generated in the fused monolayer with pipette tips after achieving 90% fusion, followed by washing twice with 1 × PBS. Cells were continuously cultured in DMEM with 2% FBS for 48 h. Inverted microscopy was used to capture 0 h and 48 h healing images of the trauma, and the relative area of trauma closure was analyzed using Image J software (Version Fiji).

### 2.8. Statistical Analysis

Most of the data analysis and visualization were carried out mainly using R software (v3.6.3), and the rest was conducted mainly using GraphPad Prism software (v8.0.0). Differences in survival between groups were determined by Kaplan–Meier analysis and the log-rank test. Subtypes, clinicopathological features, risk scores, neoantigens, TMB, MSI, immune checkpoint expression, and immune infiltration levels were determined by the Pearson correlation test. The statistical significance of differences between groups was determined by Student‘s *t*-test, and a one-way ANOVA was used for comparison between groups. A *p*-value ≤ 0.05 was considered statistically significant.

## 3. Results

### 3.1. Expression of CEP55 mRNA and Protein in Normal and Tumor Tissues

To investigate the tissue distribution of CEP55, we first analyzed the normalized expression levels of CEP55 in a variety of normal tissues using the HPA database. The results showed that CEP55 mRNA expression levels were highest in the thymus and testis (nTPM > 20) and medium in the tonsils and lymph nodes (nTPM > 10). In most other normal human tissues, CEP55 mRNA expression levels were detectable (nTPM < 10), but even lower (nTPM < 5) (Figure 1A). Next, to explore the differences in CEP55 between tumor tissues and corresponding paraneoplastic tissues, we analyzed CEP55 expression in different tumors in the TCGA dataset using the TIMER2 tool. CEP55 mRNA was significantly upregulated in 21 cancers, including bladder cancer (BLCA), breast cancer (BRCA), cervical squamous cell carcinoma and endocervical adenocarcinoma (CESC), cholangiocarcinoma (CHOL), colon adenocarcinoma (COAD), esophageal carcinoma (ESCA), glioblastoma multiforme (GBM), head and neck squamous cell carcinoma (HNSC), kidney chromophobe (KICH), kidney renal clear cell carcinoma (KIRC), kidney renal papillary cell carcinoma (KIRP), liver hepatocellular carcinoma (LIHC), lung adenocarcinoma (LUAD), lung squamous cell carcinoma (LUSC), pancreatic adenocarcinoma (PAAD), pheochromocytoma and paraganglioma (PCPG), prostate adenocarcinoma (PRAD), rectum adenocarcinoma (READ), stomach adenocarcinoma (STAD), thyroid carcinoma (THCA), and uterine corpus endometrial carcinoma (UCEC) (Figure 1B). For partial cancers lacking normal tissue, we also used the difference in CEP55 mRNA expression on GEPIA2.0 and UALCAN directly from TCGA and GTEx data. The combined results showed that there was higher expression of CEP55 mRNA in adrenocortical carcinoma (ACC), BLCA, BRCA, CESC, COAD, DLBC, GBM, HNSC, KIRC, LUAD, LUSC, ovarian serous cystadenocarcinoma (OV), PAAD, READ, SKCM, STAD, THYM, UCEC, and uterine carcinosarcoma (UCS), but lower levels in acute myeloid leukemia (LAML) (Figure 1C,D).

In addition, to assess the expression level of CEP55 at the protein level, we retrieved data from TCGA and compared the results with immunohistochemical images provided by the HPA database. As shown in Appendix A, similar analysis results were obtained for both databases. CEP55 immunohistochemistry (IHC) staining was weaker in normal breast, glial, oral (head and neck), lung, ovary, pancreas, kidney, and endothelial tissues (endometrium) and higher in tumor tissues. However, there was no significant difference in CEP55 protein levels in liver and liver tumor tissues.

### 3.2. Correlation between CEP55 Levels and Clinicopathology in Various Tumors

To explore the association between CEP55 expression in multiple cancers and clinicopathological features, we first assessed CEP55 expression in patients with stage I, II, III, and IV cancers by TCGA cancer type. We found that CEP55 expression was significantly associated with tumor stage in 15 types of cancer, including BLCA, BRCA, CESC, CHOL, COAD, ESCA, HNSC, KIRC, KIRP, LIHC, LUAD, LUSC, READ, STAD, and UCEC. Notably, in stages II and III, CEP55 levels were highly expressed in most cancers (Figure 2A). Meanwhile, CEP55 protein levels were significantly highly expressed in all III stages of BRCA, HNSC, LUAD, PAAD, KIRC, and uveal melanoma (UVM), except for PAAD and KIRC in the II stage, where CEP55 levels were significantly elevated compared to control protein levels (Figure 2B). Moreover, we evaluated the expression level of CEP55 according to the age of patients in each tumor type. The results showed that BLCA, BRCA, CESC, CHOL, COAD, ESCA, GBM, HNSC, KIRC, KIRP, LIHC, LUAD, LUSC, PAAD, READ, STAD, and UCEC patients aged > 21 years and especially >41 years had higher mRNA expression levels (Appendix A); and in BRCA, GBM, HNSC, LUAD, OV, PAAD, KIRC, and UCEC patients between 41 and 80 years had higher CEP55 protein expression levels. Interestingly, CEP55 protein levels in HNSC, PAAD, and UCEC increased progressively with age (Appendix A). The above results suggested that the expression level of CEP55 was closely related to the stage and age of most cancer patients.

**Figure 1 cells-12-02457-f001:**
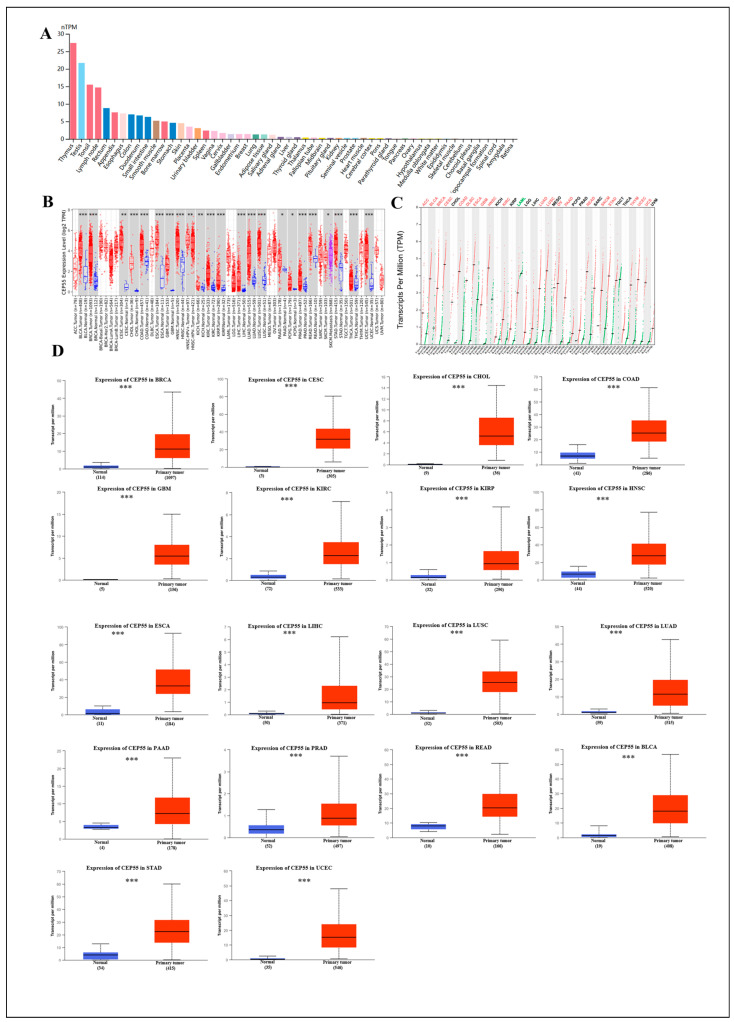
Differential expression analysis of CEP55 in normal tissues and pan-cancer. (**A**) Expression levels of CEP55 in 54 normal tissue types from the HPA database. (**B**) The expression levels of CEP55 mRNA in 33 tumor tissues and their corresponding normal tissues were analyzed on TIMER 2.0. Red and blue boxes indicate tumor tissues and normal tissues, respectively, and purple boxes represent SKCM metastatic tissues. (**C**) TCGA and GTEx data from GEPIA2.0 showed CEP55 expression levels in 27 tumors. (**D**) Expression levels were compared between normal and primary tissues of 18 cancers on UALCAN. * *p* < 0.05, ** *p* < 0.01, *** *p* < 0.001.

**Figure 2 cells-12-02457-f002:**
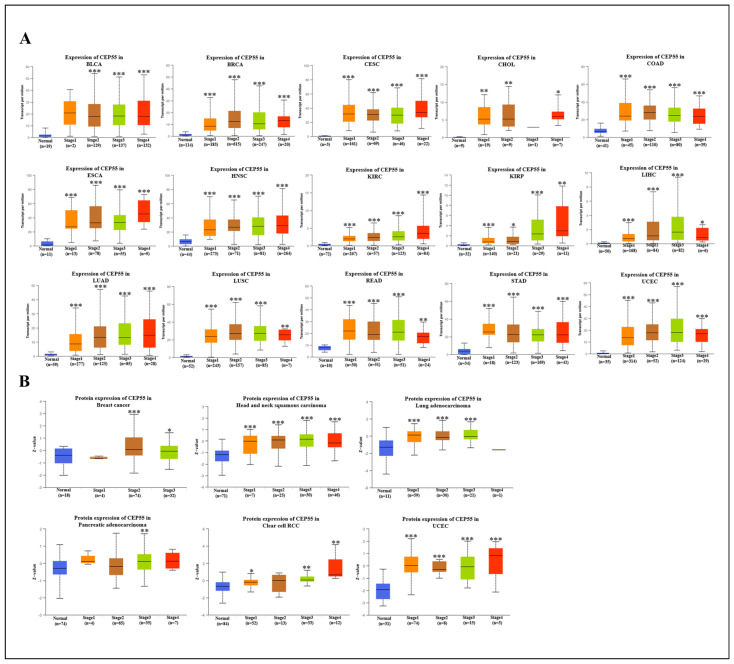
Association of CEP55 expression with four pathological cancer stages. (**A**) Impact of CEP55 mRNA levels on clinical staging in BLCA, BRCA, CESC, CHOL, COAD, ESCA, HNSC, KIRC, KIRP, LIHC, LUAD, LUSC, READ, STAD, and UCEC. (**B**) Impact of CEP55 protein levels on clinical staging in BRCA, HNSC, LUAD, PAAD, RCC, and UCEC. All data were taken from the UALCAN database. * *p* < 0.05, ** *p* < 0.01, *** *p* < 0.001 vs. normal.

### 3.3. Prognostic Value of CEP55 across Cancers

According to the TCGA database, Kaplan–Meier curves are useful to show the correlation between CEP55 expression levels and the survival of cancer patients and to assess the prognostic value of differential CEP55 expression. From the risk forest plot, we noted that CEP55 was a high-risk gene for OS in glioma (GBMLGG), pan-kidney cohort (KIPAN), brain lower grade glioma (LGG), KIRP, ACC, LIHC, KIRC, MESO, KICH, PAAD, LUAD, LAML, UVM, and THYM, while it was a low-risk gene for other cancers, particularly ESCA and CESC (Figure 3A). Plotting CEP55 in pan-cancer OS Kaplan-Meier curves by the GAPIA2.0 database showed that high levels of CEP55 had a worse prognosis in ACC, KIRC, LUAD, KIRP, LGG, PAAD, LIHC, and MESO, while a better prognosis in STAD (Figure 3B). Correspondingly, regarding the association of high CEP55 expression in pan-cancer with DFS, the forest plot showed a significant correlation in the poor prognosis of patients with GBMLGG, KIPAN, KIRP, LGG, KIRC, KICH, ACC, LIHC, MESO, PAAD, LUAD, UVM, and GBM; however, CEP55 expression exhibited the opposite relationship with prognosis in OV patients (Figure 3C). KM analysis showed that individuals with ACC, KIRC, PPAD, KIRP, LGG, SARC, LIHC, MESO, and UVM and high levels of CEP55 expression had a shorter survival time (Figure 3D). In conclusion, our results are consistent with previous summaries showing significant overexpression of CEP55 in most types of cancer [17], suggesting that CEP55 may play a potentially critical role in carcinogenesis and diagnosis.

### 3.4. Gene Stability Analysis of CEP55 in Pan-Cancer

Genomic alterations are one of the characteristics of the rapid development of cancer. To assess the genetic alteration of CEP55 in various cancers, we performed a mutational analysis using the cBioPortal database and found that CEP55 was altered in 1.3% (138/10,950) of patients with pan-cancer (Figure 4A). As shown in Figure 4B, the highest CEP55 alteration frequency of 6.24% (33/529) was present in uterine corpus endometrial carcinoma, where the most mutations were present. In addition, we also analyzed the mutation frequency of the CEP55 gene in different types of tumors, and the results showed that mutation, amplification, and deep deletion were the main types of mutation forms in the top 5 in uterine carcinosarcoma (5.26%), diffuse large B-cell lymphoma (4.17%), prostate adenocarcinoma (2.83%), and bladder urothelial carcinoma (2.43%), respectively. We also identified and visualized mutation loci in CEP55, where missense mutations were the most common type of mutation with 68, followed by truncation mutations with 16; and the D293N locus was the most frequently mutated locus, with six missense mutations occurring at this locus (Figure 4C).

Tumor formation is influenced by genetic and epigenetic modifications, and DNA methylation is one of the most important modifications in the field of epigenetics. Therefore, we investigated the correlation between CEP55 and key methylation transferases and found that DNMT1, DNMT2 (TRDMT1), DNMT3A, and DNMT3B were highly correlated with CEP55 in most tumors, especially in DLBC, LIHC, THYM, and BRCA (Figure 4D,E). Furthermore, aberrant alterations in the DNA methylation of key genes are thought to be an important cause of cancer development. We found that CEP55 was the most significant and important methylation site in cg25827255, cg25314624, and cg04026927 by analyzing the DNMIVD database (Figure 4F). To further estimate the reproducibility and validity of the selection method and diagnostic model, the ROC curve of the logistic regression model could reach 0.797, which showed good confidence (Figure 4G). Meanwhile, the DNA methylation profiles of the given CpG in tumor and normal samples were analyzed by clustering heat maps (Figure 4H). Next, we analyzed the levels of CEP55 methylation in pan-cancer and its corresponding tissues using TCGA data from the UALCAN database. The results showed that the level of CEP55 promoter methylation was significantly increased in ESCA, KIRC, KIRP, PAAD, SARC, and TGCT compared to normal tissues (Figure 4I). The effect of CEP55 methylation on the OS of various cancers was further explored through the DNMIVD database, and the results showed that CEP55 methylation levels in CHOL, PAAD, PARD, KIRC, KIRP, CESC, and LIHC were significantly and negatively correlated with patient prognosis (Figure 4J). The above results suggest that the methylation level of CEP55 may be a potential biomarker for pan-cancer diagnosis.

Phosphorylation is the most widespread form of covalent protein post-translational modification and the most important form of regulatory modification in both prokaryotes and eukaryotes. CEP55 protein phosphorylation may be inextricably linked to carcinogenesis. Therefore, we examined the phosphorylation sites of CEP55 and the changes in CEP55 phosphorylation levels between primary tumor tissues and normal tissues. The phosphorylation sites of CEP55 were mainly concentrated in S23, S428, and S436 (Appendix A). CPTAC database analysis showed that a significant increase in S23 phosphorylation of CEP55 in HNSC, S2428 phosphorylation of CEP55 in HNSC, and BRCA and S436 phosphorylation of CEP55 in HNSC was observed compared to normal tissues (Appendix A). These findings suggest that phosphorylation levels of CEP55 significantly promote partial tumor progression.

### 3.5. Analysis of Patient Survival by Alternative Splicing of Differentially Expressed CEP55

AS is an essential component of gene expression regulation, enabling cells to generate great protein diversity from a limited number of genes [18]. Defects in AS, including genetic alterations and/or the altered expression of pre-mRNA and trans-acting factors, contribute to the development of many cancers [19]. Therefore, we analyzed the ASs of CEP55 by OncoSplicing and identified 2 clinically relevant AS events, alt_5primer_51257 and alt_5primer_51252. We described mainly alt_5primer_51257 here; the other was given in Appendix A. COAD, ESCA, and STAD showed higher percent spliced-in (PSI) values in cancer samples than in normal samples (Figure 5A,B). Figure 5C summarizes the statistical results of PSI differences between tumor and normal/adjacent tissues, as well as the correlation with OS and PFI in pan-cancer. Kaplan–Meier curves with a partial prognostic value are presented in Figure 5D. The results showed that high PSI predicted higher DFI for LUAD, higher PFI for LUSC, and conversely, lower PFI for TGCT. However, high PSI was not significantly associated with OS in cancers including LUAD, LUSC, and ESCA. Overall, these results exemplify that regulating CEP55 can act as one of the important biological events affecting cancer progression.

### 3.6. Gene Enrichment and Functional Analysis of CEP55 in Pan-Cancer

To understand the functional role of CEP55 in cancer and to explore the potential mechanisms of related genes in tumorigenesis and progression, we performed functional and pathway enrichment analyses. We obtained and displayed 20 proteins interacting with CEP55 from the STRING web tool, and the associated protein networks are shown in Figure 6A. In addition, we analyzed the top 100 genes associated with CEP55 expression in 33 cancers of TCGA by GEPIA2, of which the top 5 were cyclin-A (CCNA2), cyclin dependent kinase 1 (CDK1), kinesin family member 11 (KIF11), marker of proliferation Ki-67 (MKI67), and Polo-like kinase 1 (PLK1), which were significantly positively associated with CEP55 (Figure 6B). Meanwhile, according to the heat map, CEP55 expression was positively correlated with these five genes in most tumor types, especially in ACC, BRCA, KICH, PRAD, and THYM (Figure 6C). Next, we scored the top 100 CEP55-related genes from the above cancers by gene set variation analysis (GSVA) and found significant positive correlations with apoptosis, cell cycle, and DNA damage pathways, as well as significant negative correlations with the hormone ER, RASMARK, and RTK pathways (Figure 6D). Further analysis by gene set enrichment analysis (GSEA) revealed that CEP55-related genes were significantly enriched in BRCA, KICH, and PRAD, but not in ACC and THYM (Figure 6E). GO analysis revealed CEP55 and related genes were mainly associated with cell division, cell cycle, and mitotic spindle organization in biological processes (BP), nucleus, cytosol, and midbody in cellular components (CC), and closely linked with protein binding, ATP binding, and microtubule binding in molecular functions (MF) (Figure 6F). In addition, cell cycle, oocyte meiosis, progesterone-mediated oocyte maturation, and the p53 signaling pathway were closely associated with CEP55 and related genes (Figure 6G).

**Figure 5 cells-12-02457-f005:**
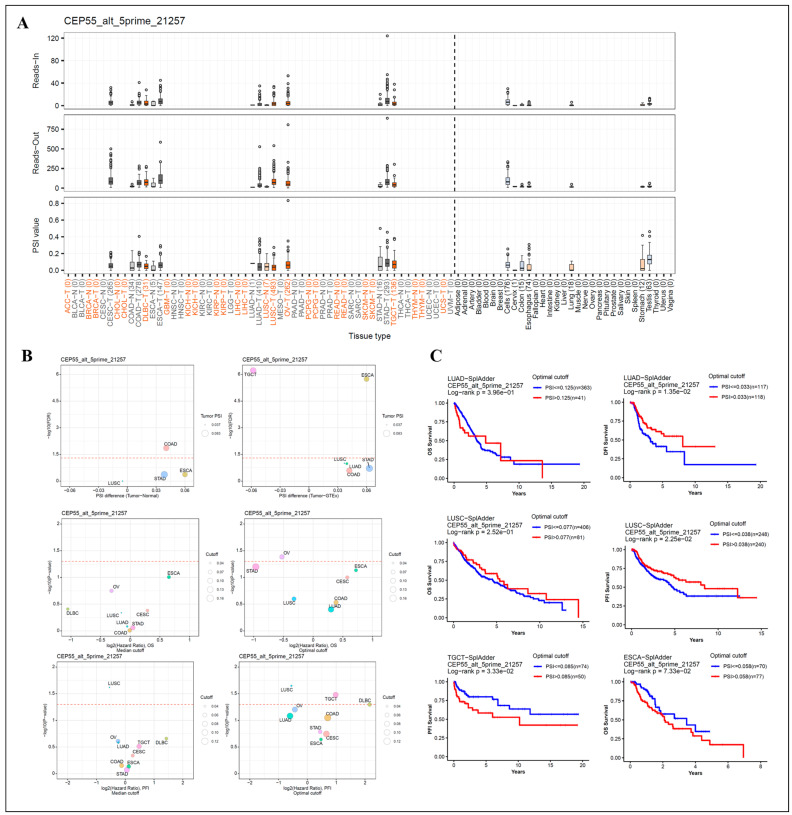
Correlation of CEP55 alternative splicing with patient prognosis. (**A**) The reads-in, reads-out, and PSI values of CEP55_alt_5primer_51257 were analyzed in pan-cancer, adjacent, and normal tissues, respectively. (**B**) Differences in PSI between tumor, adjacent normal tissue (left) and tumor, GTEx normal tissue (right), and associations with OS and PFI; the red dashed line is the FDR of 0.05, the dot size represents tumor PSI values, and different colors mark different cancers. (**C**) Kaplan–Meier curves for patient OS, PFI, or DFI prediction are shown. All data were obtained from OncoSplicing.

**Figure 6 cells-12-02457-f006:**
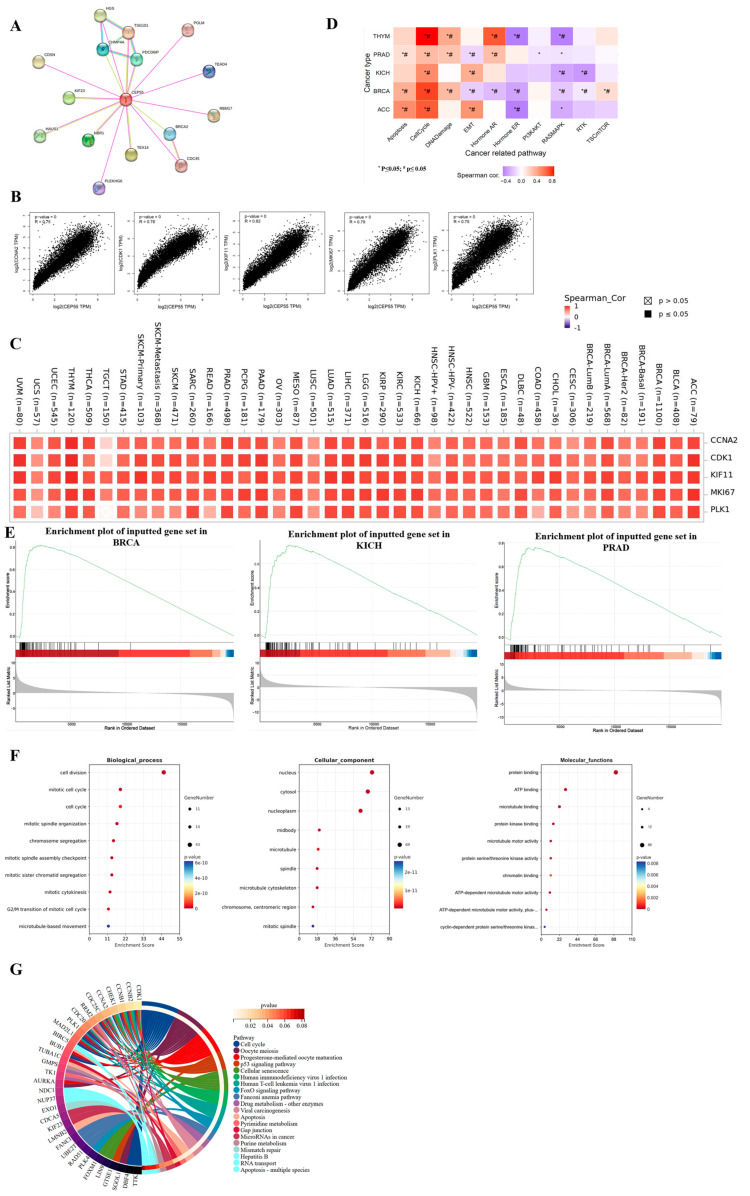
CEP55-associated gene enrichment and pathway analysis. (**A**) The STRING tool was used to identify the top 20 protein network maps associated with CEP55, where different colors represent different individual proteins. (**B**) The top 100 genes associated with CEP55 in the TCGA database were obtained using the GEPIA2 tool, and the correlation of CEP55 with the selected top 5 genes (CCNA2, CDK1, KIF11, MKI67, and PLK1) in pan-cancer was analyzed. (**C**) The heat map shows the correlation between CEP55 and the top 5 associated genes in the individual cancer types of all TCGA tumors analyzed by TIMER2. (**D**) The figure summarizes the association between the GSVA score and the activity of cancer-related pathways in selected cancers. *: *p* value ≤ 0.05; #: FDR ≤ 0.05. (**E**) GSEA enrichment analysis results. (**F**) GO enrichment bubble maps for biological process terms, cellular component terms, and molecular function terms. (**G**) A circle diagram of the top 100 gene-enriched pathways associated with CEP55. Only the key genes for each pathway are listed at the left end of the corresponding color band.

### 3.7. The Effects of CEP55 on Proliferation and Migration of Breast Cancer

To explore the molecular biological function of CEP55, we selected breast cancer for in vitro validation analysis. A search of the UALCAN database revealed that CEP55 was significantly highly expressed in breast cancer, especially in triple-negative breast cancer (TBNC) (Figure 7A). Knockdown of CEP55 in MDA-MB-231 and CAL-148 cells by transfection with siRNA (Figure 7B). Subsequently, we adopted MTT to detect the cell viability of TNBC treated with siRNA CEP55 at different time points. The cell proliferation rate was significantly reduced after CEP55 knockdown, as shown in Figure 7C. In addition, the clonality of the TNBC was significantly decreased (Figure 7D). The EdU assay showed that silencing of CEP55 reduced the proliferative capacity of the TNBC cells (Figure 7E). Moreover, migration was significantly inhibited in the CEP55 knockdown group by the transwell migration assay in MDA-MB-231 and CAL-148 cells (Figure 7F). Correspondingly, the wound healing assay in CEP55-silenced TNBC cells showed reduced cell migration (Figure 7G). In conclusion, these results illustrated that CEP55 facilitated TNBC progression by affecting proliferation and migration. Furthermore, we also performed the corresponding validation of forced overexpression of CEP55 in SK-BR-3 cells with relatively low CEP55 expression (Figure 7H–M). The results showed that the activity, proliferation, and migration of SK-BR-3 cells were significantly increased after overexpression of CEP55. Therefore, we conclude that CEP55 can significantly affect breast cancer tumor progression.

### 3.8. Regulation of Immune Cell Infiltration by CEP55 in Human Tumors

Tumor tissue does not simply contain tumor cells but also immune and inflammatory cells, tumor-associated fibroblasts, nearby mesenchymal tissue, microvasculature, and various cytokines and chemokines, which constitute a complex and integrated system of tumor microenvironment (TME) [20,21]. In recent years, more evidence suggests that the immune infiltration that promotes cancer progression in TME has become increasingly important [22]. To determine the possibility of CEP55 as a target for tumor immunotherapy, the relationship between CEP55 expression and the level of immune infiltration in pan-cancer was investigated. The relationship between CEP55 and the composition of tumor-infiltrating immune cells (TIICs) was first obtained by the TIMER2 and CIBERSORT algorithms. Our results showed that CEP55 was significantly and positively correlated with neutrophils in THYM, KIRC, PRAD, KIPAN, LGG, LIHC, LUAD, DLBC, KIRP, OV, PRAD, BRCA, SKCM, STAD, and KICH. Also, dendritic cells were positively associated with CEP55 in BLCA, BRCA, PAAD, OV, PCPG, LGG, LIHC, KIPAN, THCA, STES, STAD, THYM, LIHC, THCA, KIRC, and PRAD. Moreover, CEP55 was significantly positively correlated with B cells, T cell CD4, and T cell CD8 cells in THYM, KIRC, LGG, LIHC, PRAD, and KIPAN. Conversely, B cells, T cell CD4, and macrophages were significantly negatively correlated with CEP55 in LUSC, STES, and STAD (Figure 8A–C). Meanwhile, single-cell data analysis showed that CEP55 affected T cell proliferation in BRCA, CRC, KIRC, LIHC, and NSCLC and was highly expressed in regulatory T cells (Appendix A). Additionally, the correlation of CEP55 expression in THYM, KIPAN, KIRC, ESCA, ACC, and STES with purity was worth considering. Of note, CEP55 expression also showed a positive correlation with neoantigens in GBM, ACC, READ, and UCS, and a negative correlation with neoantigens in MESO, CHOL, TGCT, and KICH (Figure 8D).

**Figure 7 cells-12-02457-f007:**
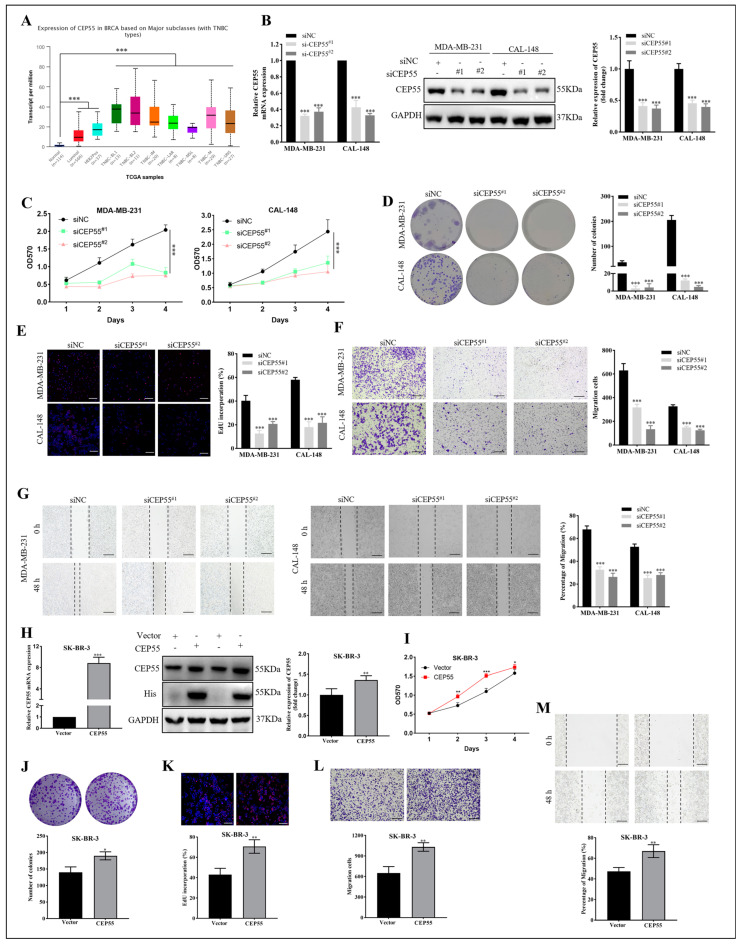
Effects of CEP55 on breast cancer cell proliferation and migration. (**A**) The expression levels of CEP55 in normal breast cancer and different subtypes of breast cancer were analyzed by the UALCAN database. (**B**) qRT-PCR and western blot were used to evaluate the efficiency of CEP55 silencing in MDA-MB-231 and CAL-148 cells. (**C**) MDA-MB-231 and CAL-148 cell proliferation was detected by MTT assays. (**D**) Colony formation assay in MDA-MB-231 and CAL-148 cells. (**E**) EdU assays in MDA-MB-231 and CAL-148 cells. (**F**) Transwell assays in MDA-MB-231 and CAL-148 cells. (**G**) Wound healing assay in MDA-MB-231 and CAL-148 cells. SK-BR-3 cells were transiently transfected 48 h with plasmids carrying CEP55 cDNA or vector only, (**H**) and then subjected to qRT-PCR and western blot analysis of CEP55 expression; (**I**) subjected to the MTT assay; (**J**) subjected to the colony formation assay; (**K**) subjected to the EdU assays; (**L**) subjected to the transwell assays; and (**M**) subjected to the wound healing assay. Data were shown as mean ± SD. * *p* < 0.05, ** *p* < 0.01, *** *p* < 0.001 vs. siNC or vector.

Next, we further analyzed the role of CEP55 in the TME by using an immune score reflecting the proportion of infiltrating immune cells in the tumor tissue, a stromal score reflecting the proportion of stromal cells in the tumor tissue, and an ESTIMATE score reflecting the tumor immune microenvironment and tumor purity [23]. We found a significant positive correlation between high CEP55 expression and immune, stromal, and ESTIMATE scores in KIPAN, KICH, and THCA (not listed), and conversely, a negative correlation in LUSC, STAD, and STES (not listed), confirming that CEP55 expression significantly regulates immune infiltration in pan-cancer (Figure 8E). There is growing evidence that the immune checkpoint (ICP) is one of the key components in tumor infiltration and immunotherapy [24]. Therefore, we downloaded the uniformly normalized pan-cancer dataset from the UCSC database to explore the relationship between CEP55 expression and ICP genes in cancer. The genes negatively correlated with CEP55 expression were just concentrated in NB and THYM. In contrast, CEP55 levels in most cancers were positively correlated with the expression of ICPs; for example, more than 30 ICP genes were positively correlated with high CEP55 expression in LIHC, OV, PAAD, UNM, LGG, THCA, KIPAN, KIRC, and PRAD. Notably, the stimulaotry gene, high mobility group box 1 (HMGB1), was significantly positively associated with CEP55 in all 33 cancers, suggesting involvement in CEP55-related ICP effects (Figure 8F). As an emerging independent predictor of the efficacy of immune checkpoint inhibitor (ICI) therapy, TMB and MSI correlate with the efficacy of multiple tumor types of ICIs alone or in combination with two ICIs and have been shown to be predictive markers of the efficacy of pan-cancer immunotherapy [25]. As shown in Figure 8G, CEP55 showed a significant positive correlation with TMB in ACC, LUAD, KICH, and STAD and a negative correlation in THYM, CHOL, and KIRP. For MSI, STAD, READ, GBM, UCS, and SARC exhibited positive correlations, while DLBC, GBM, and KIPAN exhibited negative correlations with CEP55. In summary, the critical role of CEP55 in immune infiltration and ICPs may make it a promising target for tumor immunotherapy.

### 3.9. Association of CEP55 with Immune-Related Genes

The immune system maintains the normal immune response of the body and plays an important role in preventing the invasion of various microorganisms and other foreign substances and maintaining health [26,27]. However, if the body’s immunity is too strong or too weak, it may overreact to external invaders and develop abnormal symptoms. Therefore, we explored the correlation of CEP55 with immunostimulants and immunosuppressive agents and found that CEP55 was negatively correlated with more than 25 immunostimulators in most tumors (especially CESC, ESCA, LUSC, and PAAD) and with more than 15 immunoinhibitors in most tumors (especially CESC, ESCA, LUSC, and UCS). Notably, CEP55 was most significantly associated with C10orf54, TNFRF14, TNFSF13, MICB, NT5E, ULBP1, ADORA2A, KDR, and LAG3, which may serve as potential immunomodulatory targets for CEP55 treatment (Figure 9A,B). Major histocompatibility complex (MHC) is an important molecule in the process of antigen recognition, and a growing number of studies support the use of MHC as a biomarker for ICIs [28]. Our results showed a negative correlation between MHC molecules and CEP55 in most tumors; conversely, MHC molecules in THCA, LGG, and KIRC and TAP1 and TAP2 in most tumors showed a significant positive correlation with CEP55 (Figure 9C).

The interaction of chemokines with their receptors controls the targeted migration of various immune cells in the circulatory system and between tissues and organs. It performs the tasks of removing infectious agents, promoting wound healing, and eliminating abnormal proliferating cells to maintain tissue cell homeostasis [29]. We found that CEP55 was negatively correlated with most chemokines and chemokine receptors in most tumors. Interestingly, most chemokines and chemokine receptors were significantly positively correlated with CEP55 in THCA and KIRC (Figure 9D,E). Subsequently, the levels of CEP55 were compared between immunotherapy responders and non-responders, showing that CEP55 was expressed at significantly high levels only in urothelial cancer (Figure 9F,G). In conclusion, we confirmed that CEP55 may be closely related to immune-related genes in pan-cancer and could be further explored as a target molecule.

### 3.10. Exploring the Responsiveness of CEP55 in Oncology Therapy and Potential Drugs Targeting CEP55

Targeted therapies, immunotherapies, and emerging cellular immunotherapies are emerging as a new hope for many cancer patients with better efficacy and fewer adverse effects. To explore whether CEP55 can be used as a marker of change in the course of cancer treatment response, we first retrieved the trend of CEP55 in the process of CRC and BRCA treatment by ROC Plotter. The data showed high CEP55 expression in CRC patient non-responders and a significant reduction in CEP55 levels in responders after chemotherapy, Bevacizumab, and Irinotecan interference. Meanwhile, in BRCA, responders after pharmacological intervention had lower CEP55 expression, especially those treated with endocrine therapy, who had the lowest CEP55 levels and were accompanied by a 5-year PFS predicted AUC value of 0.637 (Figure 10A).

Despite the irreplaceable role of conventional chemotherapeutic agents in oncology treatment, they are ineffective in some patients with high CEP55. We strove to appraise highly efficacious drugs targeting CEP55 therapy, thus further expanding the avenues for targeted cancer therapy. We searched for drugs associated with CEP55 expression in RNAactDrug. As shown in Figure 10B, among the top 30 drugs (FDR < 0.05), 5 small-molecule compounds, including AZD7762, bleomycin, midostaurin, afatinib, and tanespimycin, were opposite to CEP55 mRNA expression. Similarly, we searched through the cMap tool for a large number of potential compounds associated with CEP55 expression in 9 different tumor cells and showed the top 50 promising compounds (Figure 10C). Subsequently, a potential pathway of action was demonstrated by analyzing the mechanism of action (MoA) of the compounds, showing that CEP55 expression levels were associated with three compounds of the topoisomerase inhibitor and protein synthesis inhibitor types (Figure 10D).

**Figure 9 cells-12-02457-f009:**
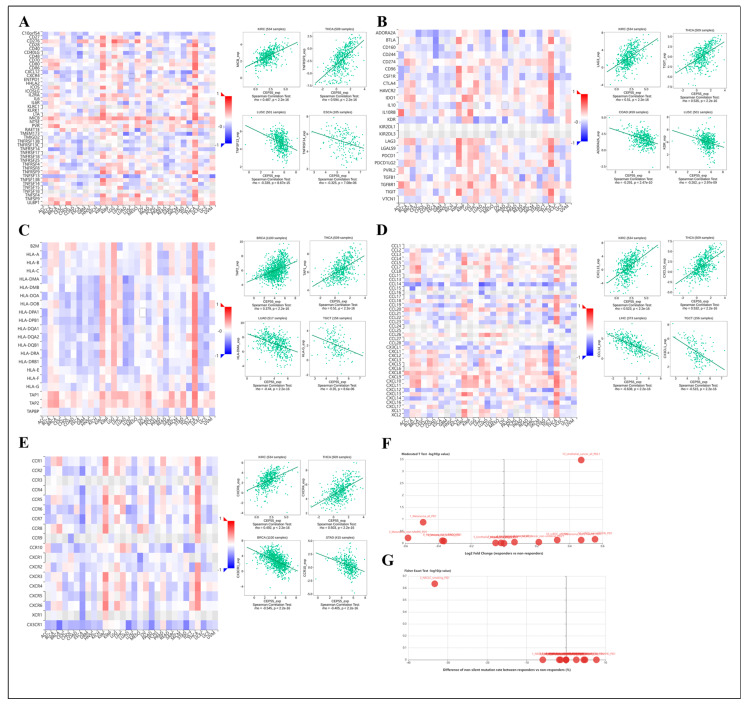
Correlation analysis of CEP55 with immune-related genes. Heat map analysis of the relationship between CEP55 expression and (**A**) immunostimulator, (**B**) immunoinhibitor, (**C**) MHC molecule, (**D**) chemokine, and (**E**) chemokine receptor in pan-cancer. The (**F**) expression difference and (**G**) mutation difference of CEP55 between immunotherapy responders and non-responders. All data were obtained from the TISIDB database.

**Figure 10 cells-12-02457-f010:**
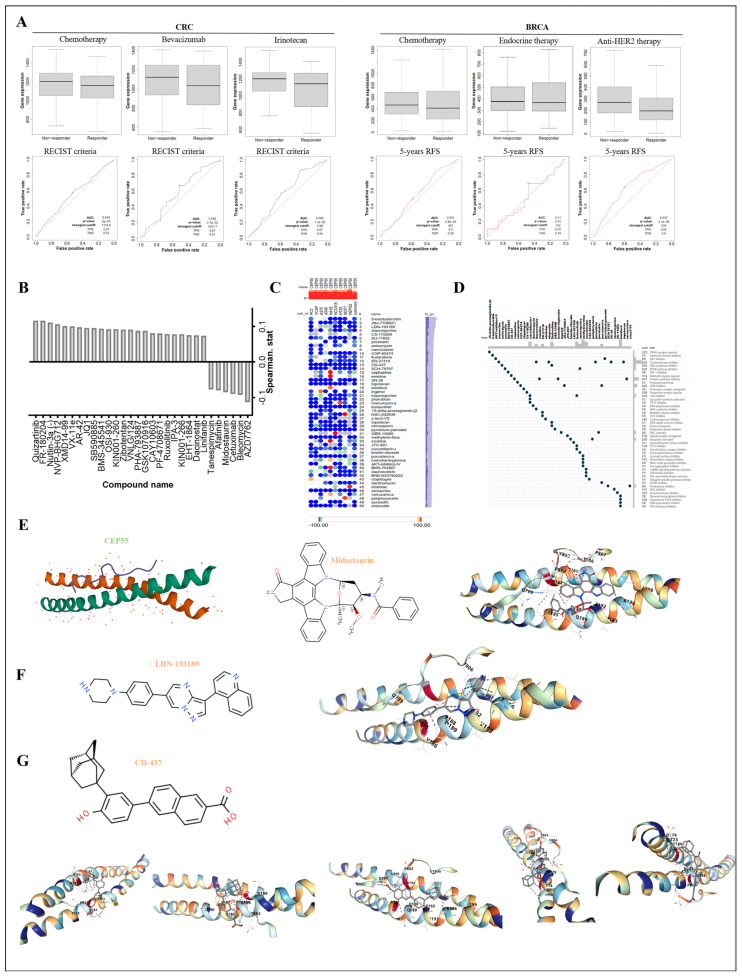
Analysis of CEP55 drug responsiveness and potential drugs targeting CEP55. (**A**) Box plots show the expression levels of CEP55 between patients after drug treatment, and ROC curves show the predictive accuracy of patient response to treatment as indicated by CEP55 levels. All data were obtained from the ROC Plotter. (**B**) Spearman’s correlation of drug-induced changes in CEP55 levels was analyzed from the RNAactDrug database, where the FDR < 0.05. (**C**) The heat map shows the top 30 possible compounds causing transcriptional alterations in CEP55 in different cells from the cMap database. (**D**) Scatter plots depict the mechanism of action (MoA) of the top 30 compounds; the count column shows the ratio of some compounds to all compounds in the cMap database with the same MoA. (**E**) (left) 3D molecular structure of CEP55 from the PDB database, (middle) molecular structure formula of midostaurin from ChemSpider, (right) molecular docking schematic of CEP55 and midostaurin, pocket: C5, vina score: −7.0, cavity volume (Å3): 69. (**F**) (left) molecular structure formula of LDN193189 from ChemSpider, (right) molecular docking schematic of CEP55 and LDN193189, pocket: C5, vina score: −7.3, cavity volume (Å3): 69. (**G**) (top) molecular structure formula of CD-437 from ChemSpider, (bottom) molecular docking schematic of CEP55 and CD-437, (left to right) pocket: C2, vina score: −8.3, cavity volume (Å3): 81; pocket: C1, vina score: −7.8, cavity volume (Å3): 105; pocket: C5, vina score: −7.4, cavity volume (Å3): 69; pocket: C4, vina score: −7.2, cavity volume (Å3): 70; pocket: C3, vina score: −7.1, cavity volume (Å3): 75.

To further explore whether these potential compounds could interact with CEP55, we performed homology modeling of the CEP55 protein and small-molecule compounds docked from RNAactDrug and cMap sources. Docking and scoring using CB-Dock2 showed significant effects of midostaurin and LDN193189 at the C5 pocket of CEP55, with vina scores of −7.0 and −7.3, respectively (Figure 10E,F). Interestingly, all pockets of CEP55 showed good interaction with CD-437, as indicated by the docking scores, with all vina scores less than −7 and up to −8.3 (Figure 10G). In conclusion, these compounds target the cellular signaling pathways of tumor cell differentiation and proliferation as therapeutic targets, providing a viable guide for the development of new antitumor targeting agents or alternatives to conventional chemotherapeutic agents.

## 4. Discussion

Currently, the increasing incidence and mortality of cancer worldwide is a major obstacle to extending life expectancy [30,31]. Therefore, the discovery and elucidation of the role of aberrantly expressed products in cancer detection and treatment are warranted. Herein, we comprehensively presented the differential expression of CEP55 in pan-cancer and normal tissues based on large-scale bioinformatics and elucidated its clinical significance. We further investigated the role of DNA methylation, AS, and gene enrichment of CEP55 in the regulation of tumor progression in human cancer. Moreover, we elaborated on the multi-omics features of CEP55 and its role in tumor immunity and screened latent target compounds. Overall, our analysis may provide a potential direction for future treatment of various tumors by targeting CEP55.

Any interference in cytokinesis during the early stages of tumor formation can lead to the appearance of aneuploidy cells, which then evolve into unstable and more carcinogenic cells. CEP55 is a mitotic phosphorylation protein that moves from the centrosome to the midbody during late mitosis and promotes shedding, significantly affecting cell proliferation and migration [32]. Clinically, CEP55 has been found to be overexpressed in many cancer types [10,33,34]. Consistent with previous studies, CEP55 expression in cancer tissues was significantly higher than its expression in healthy tissues in most of the cancers analyzed, especially in BRCA, GBM, HNSC, LUAD, PAAD, and UCEC with abnormally active CEP55 mRNA and protein levels. In addition, we revealed that high levels of CEP55 significantly affected the pathological staging of tumors in BRCA, HNSC, and LUAD, suggesting that CEP55 overexpression may be involved in tumor progression and invasion. Similarly, the levels of CEP55 were significantly elevated in patients with BRCA, HNSC, LUAD, PAAD, and UCEC tumors in the middle age group and above (>41 years). By Kaplan–Meier survival analysis, we found that abnormally high expression of CEP55 in ACC, KIRC, PPAD, KIRP, LGG, LIHC, and MESO was strongly associated with poor prognosis in OS and DFS. Our data broaden the scope of observations from previous reports [33,35,36].

Genomic alterations cause abnormal and uncontrolled cell growth, which drives most tumorigenesis [37,38]. We used the cBioPortal platform to analyze the data and found that CEP55 was mutated in all of the selected tumors. According to the results, CEP55 alterations in patients suffering from tumors are non-synonymous alterations containing mainly mutations, amplifications, and deep deletions. The frequency of genomic alterations dominated by CEP55 mutations was highest in uterine corpus endometrial carcinoma (4.91%, 26/529), followed by diffuse large B-cell lymphoma (2.08%, 1/48); and those dominated by amplification in uterine carcinosarcoma (3.51%, 2/57), followed by uterine corpus endometrial carcinoma (0.76%, 4/529); and those dominated by deep deletions in prostate adenocarcinoma (2.23%, 11/494), followed by diffuse large B-cell lymphoma (2.08%, 1/48), recommending that we should pay attention to the relationship between CEP55 mutations and tumor development.

Protein post-translational modifications (PTM) profoundly alter protein function, and their different domains and crosstalk significantly affect tumorigenesis, tumor metastasis, and tumor transformation [39,40]. In our analysis, both maintenance methylation (DNMT1, DNMT2) and de novo methylation (DNMT3a, DNMT3b) were found to be significantly associated with pan-cancer, resulting in the methylation of CEP55 at positions cg25827255, cg25314624, and cg04026927. Notably, the methylation level of CEP55 was significantly negatively correlated with the prognosis of some tumors, which was consistent with the findings of Yang et al., who demonstrated that patients with high methylation and low expression of CEP55 had a better prognosis than those with low methylation and high expression of CEP55 [41]. Therefore, increasing the methylation level of CEP55 is also a new way to target tumor therapy. To date, only a few studies have shown the role of phosphorylation and activation of CEP55 in tumors [42]. Our results have demonstrated that phosphorylation levels of S23, S428, and S436 in CEP55 were significantly elevated in HNSC and BRCA, suggesting that CEP55 phosphorylation may play a role in tumorigenesis.

In addition, our analysis explored the co-expressed genes associated with the CEP55 protein network. The results showed that CCNA2, CDK1, KIF11, MKI67, and PLK1 were significantly and positively correlated with CEP55 in pan-cancer. Moreover, CEP55 was found to have positive effects on apoptosis, cell division, cell cycle, DNA damage, and repair by KEGG and GO enrichment analyses. Functionally, loss of function of CEP55 leads to late gestational death, Meckel-like syndromes, and MARCH syndromes [43,44]. CEP55 overexpression significantly affects functional aneuploidy in multiple cancer types, promoting cell cycle progression (CCP) to enhance PRAD cell proliferation and TNBC invasiveness [45,46]. Mechanistically, CEP55 overexpression leads to cancer cell transformation, proliferation, mesenchymal transition, and invasion by directly interacting with the p110 catalytic subunit of PI3K and upregulating the PI3K/AKT pathway [47]. Similarly, CEP55 knockdown targets and inhibits forced mitosis of MEK1/2-PLK1, leading to cancer cell death [10]. In this study, we verified the biological role of CEP55 in breast cancer cells, and the results showed that inhibition of CEP55 significantly reduced the proliferation rate of TNBC cell lines MDA-MB-231 and CAL-148, further reducing the migration and invasion of TNBC cells. Conversely, overexpression of CEP55 significantly increased the proliferation and migration of SK-BR-3 cells. We believe these findings may be the basis for exploratory functional experiments with further implications for targeted interventions in CEP55 in pan-cancer.

TME is a highly structured ecosystem containing cancer cells surrounded by different non-malignant cell types, co-embedded in an altered vascularized extracellular matrix [48]. The immune cells in the TME, such as B cells, T cells, dendritic cells, macrophages, and NK cells, are closely related to the efficacy of immunotherapy [49]. The presence of TME enhances tumor cell proliferation, migration ability, and immune escape ability, which in turn promotes tumor development [50]. Therefore, we evaluated the correlation of CEP55 levels with immunity. The results showed that CEP55 expression was positively correlated with immune infiltration of B cells, T cells CD4+, T cells CD8+, macrophages, and dendritic cells, especially neutrophils in most tumors. Notably, our single-cell analysis data also confirmed a significant association of CEP55 with regulatory T cells. These results corroborate previous studies [41,51]. Tumor purity refers to the percentage of tumor cells in tumor tissue and is commonly used to assess the content of confounding tumor cells in a sample, providing a reference for predicting the prognosis and treatment outcome of patients with tumors [52]. Our results showed that CEP55 expression was positively correlated with tumor purity in most tumors. Thus, it is not difficult to understand that CEP55 overexpression is associated with poor prognosis in ACC, GBMLGG, LGG, KIRP, and PRAD. In addition, CEP55 appeared to have a positive effect on stromal scores, immune scores, and estimates in some cancer types, such as KIPAN, KIRC, and THCA, while showing a negative effect in LUSC, STAD, and STES. As a regulator of the immune system, ICP is crucial for maintaining autoimmune tolerance and regulating the duration and extent of immune responses in peripheral tissues [53,54]. Shortly, the over-expression or over-function of ICP molecules suppresses immune function and leads to low immunity in the body, which tends to promote tumor growth. Conversely, poor immunosuppressive function of ICP molecules leads to abnormal immune function in the body and causes the occurrence of various defective diseases. Our results showed that the level of CEP55 significantly affected most of the relevant immune detection site molecules in tumors, especially VEGFA, CD276, and HMGB1. We also found that CEP55 significantly affected NEO, TMB, and MSI in TME. Previous studies have reported the role of chemokines, cytokines, and MCH molecules [55,56], which are immune products that may be influenced by CEP55 to alter the tumor microenvironment. Our findings also revealed a strong association between CEP55 levels and immune-related genes in most tumor types. Collectively, these findings suggest that CEP55 may antagonize or promote tumor initiation and progression by mediating immune infiltration involved in tumor immune regulation.

Target-based drug discovery (TDD) is one of the major forms of first-in-class drug discovery [57,58,59]. This approach is based on an understanding of disease and specific mechanisms for a certain exclusive target that is highly relevant to the disease mechanism, thus targeting the design of macromolecule or small-molecule drugs. Over the past decades, this has been the most common method of new drug discovery and has given birth to a number of batches of antitumor drugs entering clinical studies [60,61,62]. Given the role of CEP55, we refined and screened potential compounds as new therapeutic strategies [63]. Our analysis from the RNAactDrug and cMap databases revealed that compounds including AZD7762, bleomycin, midostaurin, afatinib, and tanespimycin significantly reduced CEP55 mRNA levels. Further targeted docking showed that mitotalin and LDN193189 could bind CEP55. Notably, compound CD437 binds to CEP55 with more sites and a higher binding affinity. Thus, we suggest that CD437 may be a potential specific inhibitor of CEP55, and functional and mechanistic studies will be conducted in the following exploration. Briefly, this study revealed the binding mode of CEP55 to some compounds, which may provide more guidance for the design of small-molecule CEP55 inhibitors.

However, there are still some limitations to our study. Firstly, the analyses, experiments, and results of this study were obtained from different online datasets and are thus subject to systematic bias. Next, if CEP55 is used as a prognostic marker for pan-cancer analysis, more mechanisms and regulatory details still need to be explored in depth. Then, further experimental validation of CEP55 in pan-cancer in response to immune infiltration and activation of immunotherapy is needed. Finally, we need more in vivo and in vitro experiments to confirm the targeting effect and exact molecular mechanism of the screened small molecule compounds on CEP55.

Overall, we found that CEP55 was widely differentially expressed in normal and tumor tissues through a comprehensive pan-cancer analysis and was closely correlated with tumor patient stage and age, further revealing the correlation between CEP55 expression and clinical prognosis. We also found that increasing the methylation level of CEP55 significantly improved the prognosis of some tumor patients by epigenetic analysis. In addition, CEP55 expression was strongly associated with immune cell infiltration, immune-related genes, and immune checkpoints in a variety of cancers, suggesting that CEP55 may be involved in regulating the immune landscape of some tumors. Knockdown of CEP55 reduced the proliferation, invasion, and migration of breast cancer cells, thus exerting the tumorigenic effect of CEP55. Thus, we believe that CEP55 can be used as a promising biomarker to provide more focused suggestions and directions for clinical diagnosis, predicting patient prognosis, and assessing the TEM.

## Figures and Tables

**Figure 3 cells-12-02457-f003:**
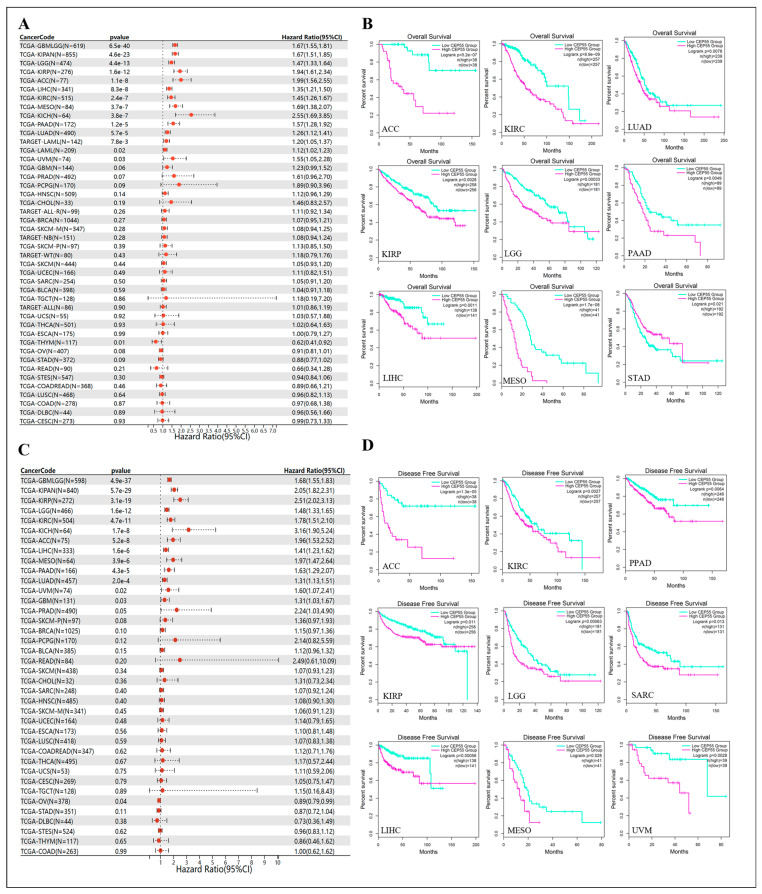
Association between CEP55 expression and overall survival (OS) and disease-free survival (DFS) in cancer patients. (**A**) Forest plot of association of CEP55 expression and OS in pan-cancer. (**B**) Kaplan–Meier analysis of the association between CEP55 expression and OS. (**C**) Forest plot of DFS associations in pan-cancer. (**D**) Kaplan–Meier analysis of the association between CEP55 expression and DFS.

**Figure 4 cells-12-02457-f004:**
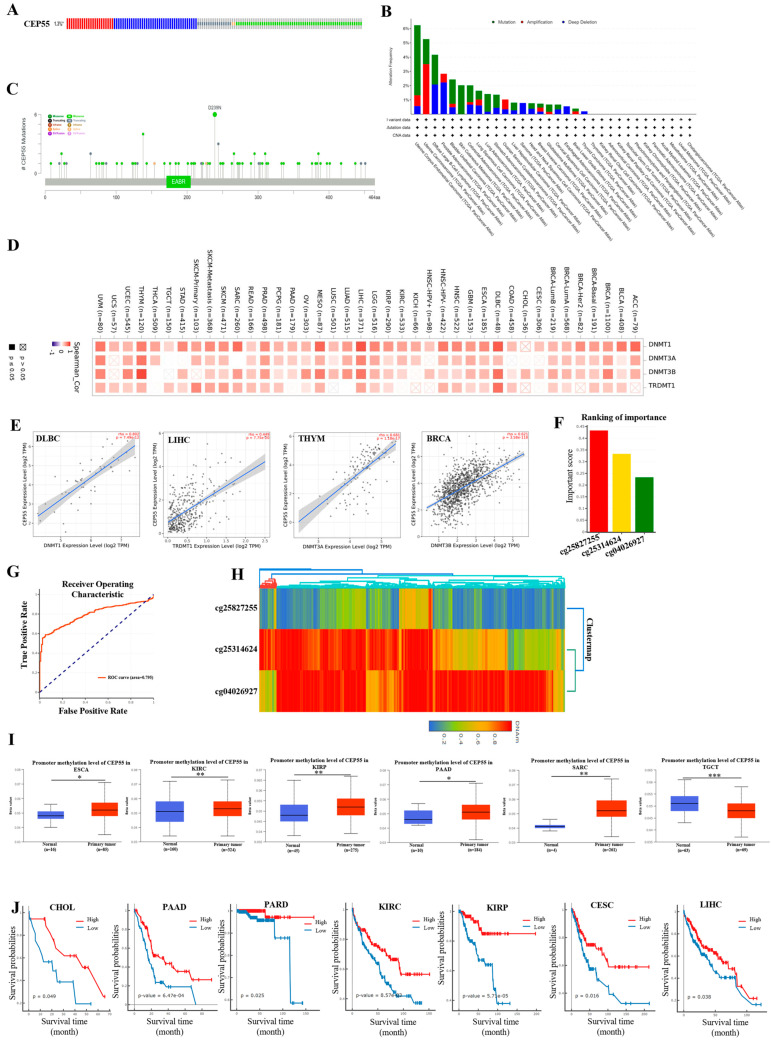
Gene stability and methylation analysis of CEP55 in pan-cancers. (**A**) cBioPhortal analysis of CEP55 mutation frequencies in pan-cancer. (**B**) Summary of CEP55 mutation frequencies in 31 TCGA cancers, including mutations, amplifications, and deep deletions. (**C**) Subtype and locus distribution of CEP55 somatic mutations. (**D**) The correlation of CEP55 with the methylation transferases DNMT1, DNMT3A, DNMT3B, and DNMT3L in pan-cancer was analyzed by TIMER2.0. (**E**) The correlation of CEP55 with DNMT1 in PRAD, DNMT3A in THYM, DNMT3B in BRCA, and DNMT3L in UVM. (**F**) The DNMIVD database was analyzed for CEP55 methylation sites (CpG Island), (**G**) logistic regression models to predict differences between tumor samples and normal samples, (**H**) and clustering heat maps of DNA methylation profiles for a given CpG in tumor and normal samples. (**I**) Promoter methylation levels of CEP55 in different tumors. (**J**) Analysis of methylation levels of CEP55 on OS in patients with different tumors. * *p* < 0.05, ** *p* < 0.01, *** *p* < 0.001 vs. normal.

**Figure 8 cells-12-02457-f008:**
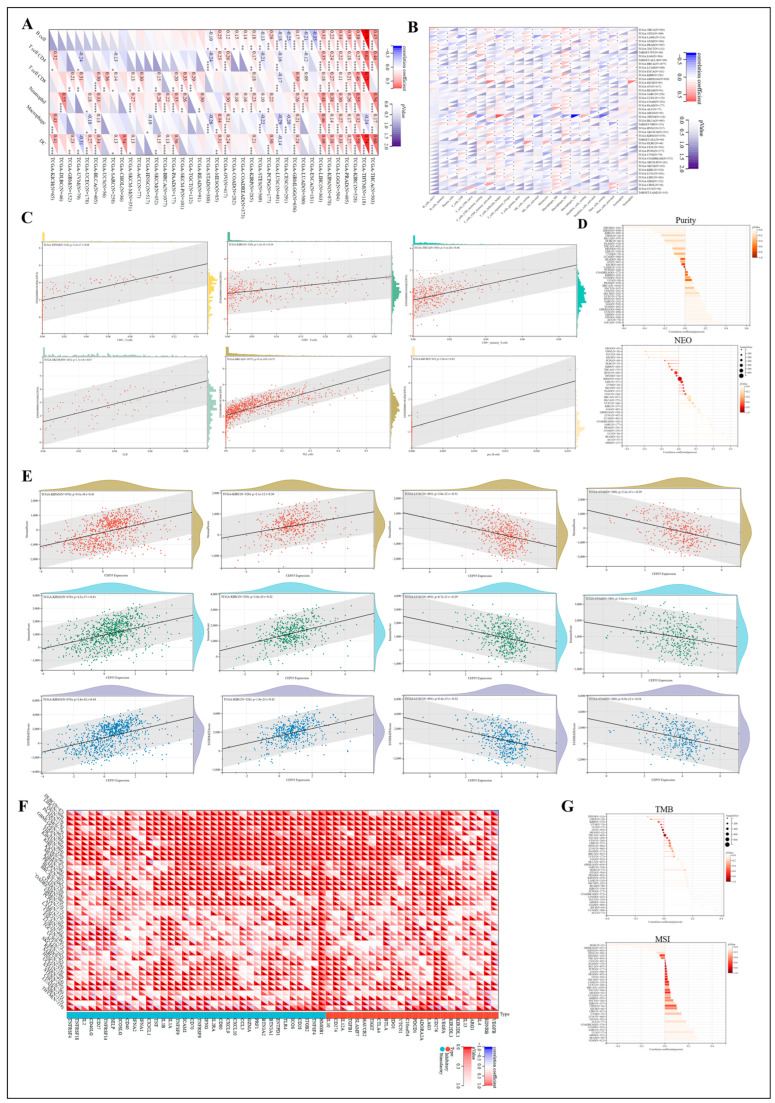
Correlation analysis of CEP55 expression with immune infiltration and immune cells in pan-cancer. Heat map showing the correlation of CEP55 expression with tumor-infiltrating immune cells (TIIC) based on (**A**) TIMER and (**B**) CIBERSORT algorithms. (**C**) Correlation analysis of CD4 T cells in THYM, CD8 T cells in KIRC, CD4 memory T cells in THYM, CLP cells in SKCM, Th2 cells in BRCA, and pro-B cells in KICH with CEP55 expression. (**D**) Correlation analysis of CEP55 with purity and tumor neoantigens (NEO). (**E**) Correlation of CEP55 expression with stromal score, immune score, and ESTIMATE score in KIPAN, KIRC, LUSC, and ATAD. (**F**) The correlations between CEP55 and immune checkpoint genes. (**G**) Correlation analysis of CEP55 with tumor mutational load (TMB) and microsatellite instability (MSI). * *p* < 0.05, ** *p* < 0.01, *** *p* < 0.001, **** *p* < 0.0001.

## Data Availability

The datasets presented in the study are included in the Section 2. Further inquiries can be directed to the corresponding authors.

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
