# Peer review of "CEP55 as a Promising Immune Intervention Marker to Regulate Tumor Progression: A Pan-Cancer Analysis with Experimental Verification"

_cells, 2023, doi:10.3390/cells12202457_

Round 1
Reviewer 1 Report
The work reported by Wang, et al looks complex and well-headed, assembling good points to reach what they are suggesting- regarding CEP55 protein as a potential target for intervention against tumor progression. Despite of, based on the literature, it seems obvious that CP55 is overexpressed in a variety of tumors, and then involved with the worst prognosis; the authors oriented a very full and complex analysis, through multiple databases, following a logical reasoning, with important questions being demonstrated along the manuscript evolution. They embraced the concern about how the CP55 is expressed in normal tissues around the organism, at the different organs, epigenetic analysis, up to how it may interact in a tumor environment and behave in a supposed drug treatment. I confess that I missed more sharpness of the figures, avoiding a deeper analysis. Probably, a large amount of the analysis and comparative data limited the space, and the PDF did not get good quality images. This point could be improved before publication. All analyses showed the CP55 marker involved as the worst scenario of the disease and with a high prevalence, promoting high levels of mitosis, highly related to immune cell infiltration, and a not good scenario of clinical prognosis closely correlated with age and severity of the disease. They clearly relate CP55 with immune response and immune tissues related to tumors, as well as, the disruption of the CP55 gene in vitro might interpose the malignancy of the cells (proliferation and infiltration). In resume, the work can bring a wide view of CP55 into a broad spectrum of cancer, guiding us to have CP55 as a promising target of intervention for cancer treatment; unfortunately, the adverse effects are still unanswered questions.
Author Response
Point by Point Response to Reviewers
We appreciate the comments and suggestions from the editor and the reviewers, which help us improve the quality of the manuscript. We have revised our manuscript accordingly. Below please find our point-by-point responses to the reviewers’ comments.
Reviewer 1
The work reported by Wang, et al looks complex and well-headed, assembling good points to reach what they are suggesting- regarding CEP55 protein as a potential target for intervention against tumor progression. Despite of, based on the literature, it seems obvious that CEP55 is overexpressed in a variety of tumors, and then involved with the worst prognosis; the authors oriented a very full and complex analysis, through multiple databases, following a logical reasoning, with important questions being demonstrated along the manuscript evolution. They embraced the concern about how the CEP55 is expressed in normal tissues around the organism, at the different organs, epigenetic analysis, up to how it may interact in a tumor environment and behave in a supposed drug treatment. I confess that I missed more sharpness of the figures, avoiding a deeper analysis. Probably, a large amount of the analysis and comparative data limited the space, and the PDF did not get good quality images. This point could be improved before publication. All analyses showed the CEP55 marker involved as the worst scenario of the disease and with a high prevalence, promoting high levels of mitosis, highly related to immune cell infiltration, and a not good scenario of clinical prognosis closely correlated with age and severity of the disease. They clearly relate CEP55 with immune response and immune tissues related to tumors, as well as, the disruption of the CEP55 gene in vitro might interpose the malignancy of the cells (proliferation and infiltration). In resume, the work can bring a wide view of CEP55 into a broad spectrum of cancer, guiding us to have CEP55 as a promising target of intervention for cancer treatment; unfortunately, the adverse effects are still unanswered questions.
Response. We thank the reviewer for making these nice comments. As suggested, we have increased the resolution of all figures for a more intuitive review (As shown our revised manuscript). Notably, there are no reports of adverse effects related to CEP55. Currently, we are continuing to explore the relevant roles and mechanisms of CEP55 in tumors and are working on the development and validation of antitumor drugs targeting CEP55, with a focus on optimizing effects and reducing associated side effects.

Reviewer 2 Report
The authors described the role of CEP55 in cancer diagnosis, prognosis and immunotherapy, providing ideas for a comprehensive understanding of CEP55 in immunotherapy and developing novel targeted therapies. In the present study, they observed the expression levels of CEP55 were significantly higher in most of the tumors than in the corresponding normal tissues, and it correlated with the pathological grade and age of the patients and affected the prognosis.
Overall I found the manuscript is well written and falls within the scope of the Journal.
Author Response
Point by Point Response to Reviewers
We appreciate the comments and suggestions from the editor and the reviewers, which help us improve the quality of the manuscript. We have revised our manuscript accordingly. Below please find our point-by-point responses to the reviewers’ comments.
Reviewer 2
The authors described the role of CEP55 in cancer diagnosis, prognosis and immunotherapy, providing ideas for a comprehensive understanding of CEP55 in immunotherapy and developing novel targeted therapies. In the present study, they observed the expression levels of CEP55 were significantly higher in most of the tumors than in the corresponding normal tissues, and it correlated with the pathological grade and age of the patients and affected the prognosis.
Overall, I found the manuscript is well written and falls within the scope of the Journal.
Response. Thanks for the positive comments.

Reviewer 3 Report
Overall, this study follows a rather standard protocol pipeline to examine the possibility of CEP55 as a both cancer progression marker or intervention target. I am convinced by the authors' arguments. I have 2 points for improvements:
1. For Section 2.1 (data collection), a list of all the sample IDs should be made available as supplementary materials. This is mandatory for data replication.
2. All figures need to be in much higher resolution.
The quality is appropriate.
Author Response
Point by Point Response to Reviewers
We appreciate the comments and suggestions from the editor and the reviewers, which help us improve the quality of the manuscript. We have revised our manuscript accordingly. Below please find our point-by-point responses to the reviewers’ comments.
Reviewer 3
Overall, this study follows a rather standard protocol pipeline to examine the possibility of CEP55 as a both cancer progression marker or intervention target. I am convinced by the authors' arguments. I have 2 points for improvements:
- For Section 2.1 (data collection), a list of all the sample IDs should be made available as supplementary materials. This is mandatory for data replication.
Response. Thanks for the comments and suggestions. According to the reviewer’s suggestion, we have now presented the sample IDs involved in the supplementary material and labeled them in the article. (As seen from lines 79 to line 80 and lines 733 to line 734 of the revised manuscript, and Supplementary Table 1.)
- All figures need to be in much higher resolution.
Response. Based on the reviewer's suggestions, we have readjusted all figures to make the better resolution. For the detail, please see our revised manuscript.
